# sSLAM: Speeded-Up Visual SLAM Mixing Artificial Markers and Temporary Keypoints

**DOI:** 10.3390/s23042210

**Published:** 2023-02-16

**Authors:** Francisco J. Romero-Ramirez, Rafael Muñoz-Salinas, Manuel J. Marín-Jiménez, Miguel Cazorla, Rafael Medina-Carnicer

**Affiliations:** 1Departamento de Informática y Análisis Numérico, Edificio Einstein, Campus de Rabanales, Universidad de Coŕdoba, 14071 Córdoba, Spain; 2Departamento de Ciencia de la Computación e Inteligencia Artificial, Universidad de Alicante, Carretera San Vicente del Raspeig s/n, 03690 San Vicente del Raspeig, Spain; 3Instituto Maimónides de Investigación en Biomedicina (IMIBIC), Avenida Menéndez Pidal s/n, 14004 Córdoba, Spain

**Keywords:** SLAM, artificial markers, marker map, localization

## Abstract

Environment landmarks are generally employed by visual SLAM (vSLAM) methods in the form of keypoints. However, these landmarks are unstable over time because they belong to areas that tend to change, e.g., shadows or moving objects. To solve this, some other authors have proposed the combination of keypoints and artificial markers distributed in the environment so as to facilitate the tracking process in the long run. Artificial markers are special elements (similar to beacons) that can be permanently placed in the environment to facilitate tracking. In any case, these systems keep a set of keypoints that is not likely to be reused, thus unnecessarily increasing the computing time required for tracking. This paper proposes a novel visual SLAM approach that efficiently combines keypoints and artificial markers, allowing for a substantial reduction in the computing time and memory required without noticeably degrading the tracking accuracy. In the first stage, our system creates a map of the environment using both keypoints and artificial markers, but once the map is created, the keypoints are removed and only the markers are kept. Thus, our map stores only long-lasting features of the environment (i.e., the markers). Then, for localization purposes, our algorithm uses the marker information along with temporary keypoints created just in the time of tracking, which are removed after a while. Since our algorithm keeps only a small subset of recent keypoints, it is faster than the state-of-the-art vSLAM approaches. The experimental results show that our proposed sSLAM compares favorably with ORB-SLAM2, ORB-SLAM3, OpenVSLAM and UcoSLAM in terms of speed, without statistically significant differences in accuracy.

## 1. Introduction

Simultaneous localization and mapping (SLAM) is the process by which a 3D map of an unknown environment is created while navigating over it [1]. This technique was initially implemented in robotics to achieve robot autonomy. In recent years, cameras have been incorporated into SLAM systems, since they are low-cost sensors, giving rise to visual SLAM (vSLAM) methods, which can efficiently solve the position estimation problem [2].

Most of the vSLAM methods use landmarks of the environment, represented as keypoints, because of their good performance [3,4]. However, keypoints are not reliable since, in many cases, they correspond to areas that may change over time, such as shadows or moving objects, making it difficult to match them over long periods. Thus, the reusability of maps generated by vSLAM methods is still an open problem. Figure 1 shows an example of two images of the same scene captured within five hours. It is clear that the amount of keypoint matches between the two images is minimal. In addition, there is a lack of texture and a degree of repetitiveness in many indoor environments, making it difficult to obtain reliable keypoints for triangulation [5].

Artificial markers placed in the environment have been proposed as a solution to the above problems [6]. Artificial markers can be created as durable hardware that can be permanently placed in key places of the environment, representing stable beacons for localization and tracking. Having at least four corners, markers can be employed to estimate the camera position reliably [7,8]. The UcoSLAM system [9] proposes combining the use of keypoints and artificial markers, taking advantage of both methods. Artificial markers are used as stable references over time, and natural markers are used to improve tracking. Although the combination of keypoints and artificial markers overcomes most of these problems, maintaining the map points can become a time-consuming task that increases proportional to the size of the environment. In addition, experiments carried out in this work show that, when SLAM is performed on a previously created map, only an average of 30% of the map points were reused. Thus, most of the points used belonged to the new video sequence (see Figure 1), degrading the performance of the system due to the management and maintenance of points that may never be reused again.

This paper proposes sSLAM, a novel visual SLAM system that efficiently combines artificial markers and keypoints to achieve a robust and fast visual localization in environments where artificial markers have been placed. Our method first creates a map of the environment by combining artificial markers and keypoints using the methodology proposed at UcoSLAM [9]. However, once the environment has been mapped, all of the keypoints are removed from the map, thus obtaining a map that only contains markers. Afterward, for localization purposes, our method requires seeing at least one marker to initialize the process. Once the initial position is known, our method fuses keypoints with marker information to achieve robust tracking. We consider that keypoints are short-term features that cannot be reliably employed in the long run but are required to obtain an accurate localization. As a consequence, we built a temporary map of keypoints that are triangulated along the sequence. However, we kept only the most recent ones instead of maintaining them all over the tracking process and disregarding older map points. As a consequence of this reduction in the number of three-dimensional points that need to be processed, our method operates faster than previous approaches. Since our method keeps less information, its tracking accuracy may be compromised. If very few map points were employed, the camera position would be poorly estimated and the chances of getting lost would increase. Therefore, the total number of keyframes maintained on the map is a parameter that can be adjusted, and thus the number of map points can be reduced to increase speed without significantly losing accuracy. The experiments conducted show that our method is faster than the state-of-the-art monocular vSLAM methods ORB-SLAM [3,10], UCOSLAM [9] and OpenVSLAM [11], without decreasing the localization accuracy.

The rest of the paper is structured as follows. In Section 2, we present the related works, while in Section 3, we provide an overview of the proposed method. In Section 4, we describe the operation of the proposed method, showing the originality of our SLAM process based on the reuse of the marker map in Section 4.2. Later, Section 5 shows the experiments performed and the results obtained by the proposed method, and, finally, in Section 6 we present the conclusions and possible future work in this topic.

## 2. Related Works

### 2.1. Visual SLAM

Visual simultaneous monocular location and mapping (vSLAM) was first tackled by the work of Davison et al. [12] and Eade and Drummond [13,14]. In both papers, the estimation of the camera position and the position of each landmark on the map are closely linked. The work PTAM by Klein and Murray [15] proposed parallelizing tracking and mapping, improving the performance of vSLAM systems in small environments. PTAM tracking is performed by the FAST [16] corner detector, where performing relocalization is impossible. Later, Mur-Artal et al. [17] developed ORB-SLAM, a SLAM system using the ORB feature detector [18], which solves the relocalization problem. The previous work is improved by the ORB-SLAM2 system [3], capable of working not only with monocular cameras but also with stereo and RGB-D cameras. Recently, the system has been extended to ORB-SLAM3 [10] which includes inertial information, as well as the creation of multiple maps, which improves the process of relocalization. In line with previous works, OpenVSLAM [11] is a vSLAM system capable of working with different camera models, such as perspective, fisheye and equirectangular.

Despite the good performance of vSLAM systems based on keypoints, they present several problems. First, relocalization is very difficult or impossible in repetitive or low-texture environments such as rooms, corridors, ceilings, etc. Second, the map’s scale is unknown when using a single camera. Finally, most of the keypoints correspond to points in the scene that may vary over time; for example, shaded areas or objects in motion.

The lack of texture in some scenarios was addressed by the authors of ORB-SLAM [5], where they propose combinubg the use of ORB keypoints, Lucas–Kanade tracking (KLT) and the fundamental matrix used to verify the geometric consistency of the track. In addition, for robust tracking during initialization, the system triangulates the points derived from the first keyframes to increase the point density of the map, extracting the ORB features and performing an epipolar search.

On the other hand, UcoSLAM [9] proposes combining the use of keypoints and square fiducial markers placed in the environment. Unlike keypoints, fiducial markers are stable marks over time, allowing them to perform robust tracking in most scenarios. In this way, UcoSLAM takes advantage of the benefits offered by fiducial markers as soon as one is detected in the scene, allowing it to obtain maps with a correct scale. In addition, the system also maintains a set of map points of the environment, which is updated as new frames are added to the system. The map points provide continuous tracking even in large scenes, and they take part in both the initialization and the optimization of the map.

The main drawback of these methods is that the number of map points increases as the environment grows, directly affecting the computation time. On the other hand, many map points were created by keypoints in areas that tend to change over time. In this paper, we propose sSLAM, a new system based on keypoints and fiducial markers. Unlike previous SLAM systems, the system uses a map of markers that have been created previously. Then, the system reuses the map by relying on scene fiducial markers and keypoints, but the latter is used temporarily, which allows for obtaining continuous tracking.

### 2.2. Fiducial Marker Systems

Fiducial markers are a widely known tool for camera pose estimation, helping to improve the accuracy of the systems, in addition to being robust and easy to detect in the environment. Furthermore, these systems solve the above problems associated with the use of keypoints.

Among all of the various forms of markers, square fiducial markers (see Figure 2b) have become the most popular [19,20,21,22]. These markers consist of a black outer border and an internal (primarily binary) encoding that uniquely identifies each marker. The main advantage of these markers is that they provide four points (their corners) that can be used to estimate the position of the marker.

Artoolkit [21] is one of the first systems based on square fiducial markers. Despite its success, it had problems with illumination changes and a high false positive rate. Its successor, Artoolkit Plus [23], solved these problems.

Nowadays, ArUco [8] and AprilTag3 [7] are the most widely used fiducial-marker-based systems. Both systems present a fast and robust method for marker detection that uses an efficient thresholding method and performs binary code error correction.

Currently, most of the methods present markers with a specific design. VuMarks is a commercial application developed by the company Vuforia (https://library.vuforia.com/, accessed on 8 February 2023), which allows for the design of customized markers, as well as their detection and tracking. Although the program is available for use, it is impossible to access the project’s source code or technical information. To solve the above issue, a new method for the creation of custom markers based on templates was proposed in [24], as well as a new method for marker detection and tracking even with occlusion.

The SPM-SLAM [6] system proposes a method for creating a map of planar markers and localizing the camera pose from a video sequence. In addition, the method deals with the ambiguity problem produced by the use of planar markers [25], avoiding incorrect camera pose estimation during the SLAM process, a problem that is also addressed in UcoSLAM [9].

Extending the methodology proposed by UcoSLAM [9], the work of Ortiz-Fernandez et al. [26] introduces the concept of the smart marker, composed of a square fiducial marker and a pose measurement system (PSM) unit. Smart markers distributed in the environment are used during the mapping process. The camera position estimation is obtained from the marker detections using image and orientation/distance measurements acquired from the PMS unit.

Finally, in order to be able to detect any fiducial marker, in this paper, we integrated within the project libraries a simplified version for the detection of custom markers based on the work [24].

## 3. System Overview

This section aims to provide a general overview of our proposal’s main elements and their mathematical notation.

### 3.1. Markers

Let us first define a marker as the tuple M={V,B}, which represents a polygon of *n* vertices that we will denote as
(1)$$M_{V}=\left\{v_{1},\ldots ,v_{n}|\forall n\geq 4,v_{i}\in R^{3}\right\},$$
where $v_{i}$ are the corner coordinates relative to the marker center. Please note that, throughout this paper, we use subindices to refer to a particular element of a tuple, e.g., $M_{V}$ is the polygon that corresponds to the marker *M*.

Our markers follow the idea proposed by Jumarker [24] (see Figure 3). By providing a template defining the marker boundary and the regions that contain bit information, one can automatically generate a set of uniquely identifiable markers, all with very similar appearances; Figure 3.

In order for *M* to be easily detected in the environment, their edges should have a high contrast against the background on which it is placed. For example, if the edge of the marker is to be fixed on a white wall, the ideal color of the edge will be black. *M* contains a set of colored regions that encode binary information. Let us denote the marker bits as
(2)$$M_{B}=\left(b_{1},\ldots ,b_{n},b_{n+1},\ldots ,b_{l}\right),$$
where $b_{i}\in \left\{0,1\right\}$, $b_{i}|1\leq i\leq n$ are the bits employed for identification (each marker has a unique ID), whereas $b_{i}|n+1\leq i\leq l$ are the bits used for cyclic redundancy checking (CRC) [27]. It is convenient to have as many CRC bits (l−n) as possible to prevent false positives. However, it reduces the number of bits for identification *n* and the total number of different markers. For example, if a marker has 24 bits in total (l=24), using 16 for CRC (l−n=16) leaves only 8 bits for identification (n=8), thus obtaining a total of $2^{8}$ different markers. Each possible state of the marker bits (0 or 1) is represented by a single color selected in the marker template at design time.

Figure 2a,c show the markers used in this work.

### 3.2. Frames

Let us denote a frame by the tuple
(3)f={θ,I,K,M},
where θ∈SE(3) is the camera pose at the instant the image *I* was captured. $I=\{\left\{I_{j}\right\},j\in \left\{1,\ldots ,n\right\}\}$ is a pyramid of images obtained by subsampling the image *I* using a scale factor σ∈(1,∞), from which, a set of keypoints K were extracted using a keypoint detector and a feature extractor. Finally, M={m} is the set of markers detected in the image *I*.

Let us define a marker observation as
(4)m={ϕ,M,W},
where *W* is the set of vertices of the marker
(5)$$W=\{w_{1},\ldots ,w_{n}|\forall n\geq 4,w_{i}=\left(x_{i},y_{i}\right)\in R^{2}\},$$
where the elements $w_{i}$ are the pixel coordinates of the vertices of the marker *M* that were observed in the image *I* and ϕ∈SE(3) is the marker pose that moves points from the marker reference system (mrs) to the global reference system (grs). Note that a reference system describes objects’ relative positions (location and orientation) in a specific space.

Please note that only the frames relevant to the SLAM process are stored in the map. These are called keyframes. In addition, note that the frame pose $f_{\theta }$ can be estimated from either keypoints or markers or both, as explained below.

### 3.3. Keypoints

Let us define a keypoint *k* as the tuple
(6)k={l,p,d},
where l∈{0,…,η} is the level of the pyramid I where it was detected, and $p\in R^{2}$ are its image coordinates. In addition, *d* is the binary descriptor of the keypoint, given by the vector
(7)$$d=\{\left(d_{1},\ldots .,d_{u}\right)|d_{j}\in \left\{0,1\right\}\}.$$

### 3.4. Map

We define the system map as
(8)$$\Upsilon =\{G,S^{M},F^{\tau },P\},$$
where $F^{\tau }$ is the set of keyframes. A key aspect of our approach is that the number of keyframes in the map is a parameter τ that can be adjusted to increase the computational speed. $S^{M}$ represents the set of all markers detected in the video sequence, and G is a connected graph that holds the connections between keyframes, where each edge of the graph represents a weight that determines the strength of the connection. The weight is determined by the number of common map points and markers observed by the two interconnected keyframes.

The set P={p} represents the map points, obtained by keypoint triangulation. We define a map point p∈P as the tuple
(9)$$p=\{x,v,\hat{d}\},$$
where $x\in R^{3}$ represents the coordinates of the point in the grs, *v* is its normal vector and $\hat{d}$ is the descriptor that best represents the point.

If the point *p* is observed by several keyframes, there are *n* keypoints that can be used to estimate its descriptor. Denote by $K^{p}$ this set; then, we consider $d_{i}$ as a descriptor $\hat{d}$ of *p* that minimizes the Hamming distance $D_{h}$ to the rest of descriptors of the set $K^{p}$:(10)$$p_{\hat{d}}=\underset{d_{j},d_{i}\in K^{p}}{\mathrm{argmin}}\sum_{j=1}^{n}D_{h}\left(d_{j},d_{i}\right).$$

The viewing direction *v* is a vector indicating the average perpendicular direction of the keyframes observing the point. If $f^{i}$ are the keyframes observing the point *p*, then
(11)$$v=\frac{\sum_{i}f_{\theta_{3}^{-1}}^{i}}{\left|\right|\sum_{i}f_{\theta_{3}^{-1}}^{i}{\left|\right|}_{2}^{2}},$$
where $f_{\theta_{3}^{-1}}^{i}$ is the third column vector from the inverse transform matrix $\theta_{3}^{-1}$ of keyframe $f^{i}$, i.e., a vector in the *z* direction of the keyframe. Thus, *v* is the normalized average of such vectors.

### 3.5. Reprojection Error

Let us define the projection of a 3D map point $p_{x}\in R^{3}$ on a frame image by the function
(12)$$\Psi (\theta ,p_{x},\lambda ),$$
where θ∈SE(3) is the transform that moves the coordinates $p_{x}$ from the grs to the frame reference system (frs) and λ represents the intrinsic parameters of the camera. Then, the projection error of $p_{x}$ with respect to its observation $o\left(p_{x}\right)\in R^{2}$ on the image is given by
(13)$$e\left(\theta ,p_{x},\lambda ,o\left(p_{x}\right)\right)=\Psi \left(\theta ,p_{x},\lambda \right)-o\left(p_{x}\right).$$

The pose of a marker *m* with regard to a camera can be estimated by minimizing the reprojection error of the marker vertices observed in the image. Since all marker corners belong to the same plane, this non-linear function can be solved using methods such as the infinitesimal plane-based pose estimation (IPPE) [25]. Nevertheless, due to noise in the localization of the corners, two solutions arise, and, in some cases, it is impossible to distinguish the correct one. This problem is known as the planar pose ambiguity and must be considered when designing the solution as explained in [9].

The reprojection error of the markers $\Gamma_{S^{M}}^{f}$ detected in a frame *f* is calculated as:(14)$$\zeta \left(\Gamma_{S^{M}}^{f},f_{\theta },\lambda \right)=\sum_{m\in \Gamma_{S^{M}}^{f}}||e\left(f_{\theta }\cdot m_{\phi },m_{M_{V_{i}}},\lambda ,m_{W_{i}}\right){\left|\right|}_{2}^{2},$$
where $f_{\theta }\cdot m_{\phi }$ transforms the marker vertices from the grs to the frs. Note that $m_{W_{i}}=\;o\left(m_{M_{V_{i}}}\right)$.

In addition, let us define the set of map points observed from frame *f* as
(15)$$\Gamma_{P}^{f}=\left\{p,k\right\},$$
where p∈P is the map point that projects onto the keypoint *k* of frame *f*. We shall define the reprojection error of the map points in frame *f* as
(16)$$\zeta \left(\Gamma_{P}^{f},f_{\theta },\lambda \right)=\sum_{\left(p,k\right)\in \Gamma_{P}^{f}}H_{p}e\left(f_{\theta },p,\lambda ,k\right)\Omega_{k}e{\left(f_{\theta },p,\lambda ,k\right)}^{T},$$
where $H_{p}$ is the Huber loss function used to reduce the influence of outliers, $f_{\theta }$ represents the pose of frame *f* and $\Omega_{k}$ is the information matrix that modulates the error in the estimation. It is defined as
(17)$$\Omega_{k}=\sigma^{k_{l}}I_{2},$$
where $I_{2}$ is the 2×2 identity matrix, and $\sigma^{k_{l}}$ weights the importance of a keypoint according to the pyramid level at which it was detected. Points found at a higher level of the pyramid have a higher weight because they are detected with higher precision, i.e., in larger images.

Finally, the set of observed map points $\Gamma_{P}^{f^{i}}$ and the vertices of the detected markers $f_{S^{M}}^{i}$ in frame $f^{i}$ are used to obtain its pose by minimizing their reprojection errors:(18)$$f_{\theta }^{i}=\underset{\theta }{\mathrm{argmin}}\left(w^{p}\zeta \left(\Gamma_{P}^{f^{i}},\theta ,\lambda \right)+w^{m}\zeta \left(\Gamma_{S^{M}}^{f^{i}},\theta ,\lambda \right)\right).$$

Since the number of map points is usually much larger than the number of observed markers, it is necessary to balance the reprojection errors of both components. To achieve this, the reprojection error of points is normalized by the factor $w^{p}$, which depends on the total number of points observed in the frame $f_{i}$:(19)$$w^{p}=\frac{1}{|\Gamma_{P}^{f^{i}}|}.$$

Likewise, the marker reprojection error is normalized by the factor $w^{m}$, which depends on the total number of vertices belonging to the markers observed in the same frame:(20)$$w^{m}=\frac{\beta_{\Gamma_{S^{M}}^{f_{i}}}}{|\Gamma_{S^{M}}^{f_{i}}|}.$$

In addition, the reprojection error of the markers is also modulated using the relative weight $\beta_{\Gamma_{S^{M}}^{f_{i}}}$. It is calculated according to the region occupied by the markers in the image based on the following idea. If a marker is detected close to the camera, it occupies a large region in the image, and the pose estimate obtained is reliable. However, as the camera moves away from the marker, the area occupied becomes smaller and the pose estimate loses reliability. Extending this principle, we make the weight of the markers directly related to the area of the convex polygon formed by the corners of the $n_{m}^{f_{i}}$ markers detected in the frame, which we denote as $W^{f_{i}}$. Then, the weight $\beta_{\Gamma_{S^{M}}^{f_{i}}}$ is given by the following equation:(21)$$\beta_{\Gamma_{S^{M}}^{f_{i}}}=\left\{\begin{array}{cc} 0 & n_{m}^{f_{i}}=0\\ \alpha_{w} & W^{f_{i}}<\alpha_{m}\\ \left(1-\alpha_{w}\right)\frac{\left(W^{f_{i}}-\alpha_{m}\right)}{\left(1-\alpha_{m}\right)}+\alpha_{w} & otherwise. \end{array}\right.$$

If at least one marker is detected in the frame $f_{i}$, the weight of the markers $\beta_{\Gamma_{S^{M}}^{f_{i}}}$ takes values between the minimum value $\alpha_{w}\in \left(0,1\right]$ and the maximum value 1, according to the marker observed area in the image $W^{f_{i}}\in \left[0,1\right]$. In addition, the weight is modulated by the parameter $\alpha_{m}\in \left[0,1\right]$, which determines the minimum observed area of the marker. Thus, when the observed area is smaller than $\alpha_{m}$, the weight of the markers is $\alpha_{w}$, and the weight obtains its maximum value when the observed markers cover the whole image.

## 4. Proposed Method

This section explains our SLAM proposal. In the first phase, (Section 4.1) of our method creates a map of the environment by combining keypoints and markers following the philosophy proposed in UcoSLAM [9]. The system initializes an empty map with the first detected keypoints and fiducial markers from a video sequence of the area to be mapped. It then continuously performs localization and mapping by adding new keypoints and markers to the map. Once the map is finished and globally optimized, all map points are discarded, and only marker information is retained.

In the second phase (Section 4.2), the method uses the map to localize the camera while moving in the same environment. While markers are stable elements that do not change their appearance (nor their position) over time, they are scarce. On the other hand, keypoints are a great tool for obtaining 3D points via triangulation that can be employed to perform localization in the short term. Therefore, in the second phase, we created temporary map points, as in the previous phase, which were combined with the existing markers. The main difference with regard to other approaches is that, in our system, the total number of keyframes (i.e., keypoints) is limited to only the last τ ones. We are assuming that older keyframes correspond to areas that we have left behind; thus, keeping them in memory only increases the computational time unnecessarily. If the system were to revisit that area later, it would create the necessary map points again. One may argue that this strategy could lead to an increase in error due to drifting. After all, having more keypoints improves localization and thus reduces drift. However, our method relies on the markers, which allow for resetting the drift whenever they are spotted. As a consequence, our method can speed up the process without a significant loss of accuracy.

Figure 4 shows the different stages of our system for the two phases mentioned above: first, the creation of the base map (Section 4.1), and then, localization on the map using a new video sequence (Section 4.2).

### 4.1. Map Creation

At the beginning, the system Υ=⌀ is initialized using either keypoints or markers. Our method first seeks a solution using keypoints and, if it is not valid, then tries to initialize with markers.

The method proposed in [3,9] is applied for keypoint-based initialization. Initially, the first two frames of the sequence, $f^{1}$ and $f^{2}$, are analyzed and the set of matches is extracted using their respective keypoint descriptors. Then, the matches are used to calculate the homography and essential matrices between both frames. The keypoints are triangulated and their reprojections are analyzed to detect if the solution is correct. If the initialization process fails, $f^{2}$ is replaced by the subsequent frames of the sequence $f^{3},f^{4},\ldots$, and the analysis of the matrices is carried out again.

The method proposed in [6,9] is employed for marker-based initialization. Given a single frame $f^{i}$, the system is able to initialize if at least one marker is detected without ambiguity. In the case of ambiguity of all detected markers, marker observations in two consecutive frames $f^{i}$ and $f^{i+1}$ are necessary for finding a good initial pose.

After the initialization process is completed, the frames used for initialization are added as keyframes to the system. The triangulated points are added as new map points to the system, and the descriptors that best represent the keypoints become part of the Bag of Words (BoW) [28] in order to be able to perform relocalization in case the system gets lost during this first phase. Finally, the system graph is initialized with the new elements introduced and their weights are updated.

Once the system has been initialized, the tracking process follows the methodology proposed by [9]. The camera position is estimated for each sequence frame using the pose obtained from the previous frame. Camera pose estimation consists of minimizing the reprojection error of the observed map points and markers; Equation (Equation 18).

When the sequence is finished, the map is stored in order to be reused later. Unlike the methods [9,11], the system removes all map keypoints P=⌀. In addition, G is updated so that the graph only contains connections between keyframes where there is at least one marker. In this way, we exclusively keep the information provided by the markers, i.e., their observations and their positions in the current map. Thus, the final state of the system will be $\Upsilon =\{G,S^{M},F^{\tau },P=⌀\}$, where $F^{\tau }$ is the set of keyframes where there is at least one marker observation. Please note that, during the process of creating the marker map and following the methodology proposed by the works [9], no limitations were set on the maximum number of keyframes that can be added to the map, i.e., τ=∞.

### 4.2. Camera Localization

Once a map of the markers of the environment is available, we can use it to localize a camera in it. An advantage of our method is that the map does not contain any camera-dependent information (only the positions of the markers). Thus, our method could be employed to localize a camera different from the one employed to create the map. This is not possible for current state-of-the-art SLAM methods since their maps are camera-dependent.

Since the system does not know the camera pose in the first frame, an initial localization using the markers $\Upsilon_{S}^{M}$ is required (Section 4.2.1). Once an initial pose is found, the method operates as a regular SLAM, creating a 3D map by the triangulation of keypoints. However, the main difference is that our method is continuously removing keyframes and points along the way, keeping only a small subset of τ keyframes (Section 4.2.3). In this way, our system drastically reduces the amount of information required to be processed to achieve a good accuracy. In addition, whenever a marker is observed along the way, the system removes the drift that keypoint-based localization may have introduced.

The following sections explain the localization process proposed in detail.

#### 4.2.1. Relocalization with Markers

We start from a system in which we have the markers $\Upsilon_{S^{M}}$, i.e., their locations on the map, the keyframes in which they were observed $\Upsilon_{F^{\tau }}$ and their interconnections $\Upsilon_{G}$. First, the system tries to perform relocalization using the positions of the first observed markers. Thus, if, in the current frame $f_{i}$, at least one marker $m\in \Upsilon_{S^{M}}$ has been correctly detected, the frame pose is estimated. If the estimated pose is invalid or there is ambiguity, the method discards it and tries again with the next frame where a marker is successfully detected. Once we have a good initial pose for frame $f^{i}$, it is used as the reference frame, and the following frames ($f^{i+1},f^{i+2},\ldots$) are used to start the mapping process. First, a keypoint search is performed on two frames ($f^{i}$ and $f^{i+1}$), restricting the search to the epipolar line. If there are a minimum number of α matches to triangulate with, both frames are added as new keyframes, and the new points are added to the map. Suppose that the number of matches is lower than specified or triangulation cannot be performed. In that case, the same process is used using the reference frame $f^{i}$ and the following frames of the sequence (($f^{i}$, $f^{i+2}$), ($f^{i}$, $f^{i+3}$), …) until the initialization process can be completed.

#### 4.2.2. Tracking

Estimating the frame pose $f^{i}$ can be defined as the problem of minimizing the sum of the projection errors of the map points and markers observed by the frame; Section 3.5.

First, let us define the reference frame $\hat{f^{i}}$ as the keyframe that has the largest number of matches with frame $f^{i-1}$, i.e., the previous frame. This allows us to perform a quick search for matches since the points are likely to appear again in the current frame. These matches, together with the vertices of the observed markers, allow us to obtain an initial estimate of $f^{i}$. Then, each map point from the set of matches is projected onto the image *I* (frame $f^{i}$). Finally, we search for the keypoints of the current frame that present a Hamming distance closest to the map point descriptor, and that are within a search radius *r*.

#### 4.2.3. Inserting and Removing Keyframes

One of the cornerstones of our method is the strategy followed for inserting and deleting keyframes. The insertion of new keyframes allows us to add new information to the map in order to have the least error tracking possible while the environment is explored. Nevertheless, it is crucial not to add more points than necessary, since they do not provide relevant information to the map and they increase the computational burden.

A frame is added to the map whenever it presents a valid pose and the number of map points matched with the current frame is below a percentage $\tau_{k}$ of the number of map points detected in the reference frame. In case the frame has at least one marker, both the frame and the marker are incorporated into the map. In addition, the system analyzes the new points to be incorporated into the map. Possible correspondences with neighboring keyframes are searched for each keypoint detected in the current frame. Following a strategy similar to the works [3,9], the map points are subject to a maintenance process so that points are kept if they are seen again in at least two thirds of the subsequent images.

Unlike the map creation process explained earlier (Section 4.1), where the number of keyframes is not limited (τ = *∞*), and also unlike the way other SLAM methods build the map, our method performs SLAM by building a map where the maximum number of keyframes is limited by the parameter τ. When a new keyframe is added to the current map, the system checks the total number of keyframes, and if it exceeds the value of τ, the oldest keyframe is removed, as well as associated map points observations. In this way, we use both keypoints and fiducial markers in order to have high precision in estimating the camera position as the system explores the scene, while forgetting the keypoints of areas that we have already visited.

## 5. Experiments and Results

This section shows the experiments conducted to validate our method, sSLAM, which was compared to the state-of-the-art monocular SLAM methods ORB-SLAM2 [3], ORB-SLAM3 [10], OpenVSLAM [11] and UcoSLAM [9]. The source code and datasets employed in our experiments are available for other researchers to reproduce our experiments (https://www.uco.es/investiga/grupos/ava/portfolio/sslam/, accessed on 8 February 2023).

This section is organized as follows. Section 5.1 and Section 5.2 analyze the different methods in indoor environments, which allow us to measure the performance when there is little texture in the environment to produce reliable keypoints. Then, Section 5.3 evaluates the methods in outdoor environments. Note that, unlike the ORB-SLAM2, ORB-SLAM3 and OpenVSLAM methods, which only work with scene keypoints, UcoSLAM and our method are able to work with fiducial markers as well. In the experiments, we employed different types of customized markers (Figure 2), and we adapted UcoSLAM so that it is able to detect them as well (instead of only ArUco markers).

For the experiments, 22 video sequences were used from different zones. The camera captured the sequences while moving between 4 and 6 km/h in each environment. The first 6 sequences belonged to the SPM dataset [6] and were recorded indoors, spotting at walls and ceilings. The remaining 16 sequences include 2 from the corridors and lobby of a building and 14 from various outdoor locations on our university campus; see Table 1. Then, the following methodology was employed for each method. First, a map of the environment (*base map*) was built with a video sequence. Second, the rest of the sequences of the same environment were employed for tracking evaluation using this base map. The process was repeated swapping the sequence employed to build the base map, thus employing a cross-evaluation approach. Note that, whereas the different vSLAM systems reuse the base map to continue the construction of the map using keypoints, or the combination of keypoints and fiducial markers (in the case of UcoSLAM), the proposed method uses only a base map composed of artificial markers, removing from the base map the keypoints that were created during the first sequence.

All experiments were executed with an Intel(R) Xeon(R) CPU E5-2630 v4 @ 2.20 GHz, with 46 GB of RAM and Ubuntu 20.04.2. Note that, for the execution of the experiments, the parameters shown in Table 2 were used for our method.

### 5.1. Indoor Evaluation in Small Areas

In this experiment, we employed the SPM [6] dataset, which consists of six video sequences recorded inside a room in which 0.16 m ArUco markers are uniformly distributed in the ceilings and walls of a room. The video sequences were recorded with a PtGrey FLEA3 camera with 1920×1080 pixel resolution, and the ground truth camera position was provided by an OptiTrack motion capture system (Figure 5c,f).

A base map (for each video sequence and method) was generated as a preliminary step in the evaluation. Then, the base maps were reused with the rest of the videos in the dataset. Unlike ORB-SLAM2, ORB-SLAM3, OpenVSLAM and UcoSLAM methods, where the map grows as the camera moves in the environment (incorporating new information to the base map), sSLAM keeps only the information of the last τ keyframes added to the map, ignoring the rest of the information; Figure 6.

To evaluate the accuracy of the different methods working with a base map (using the second video sequence), the absolute trajectory error (ATE) and the percentage of frames in which a valid pose is obtained (pTrck) were measured. The ATE was calculated as the RMSE between the camera positions for each frame and the ground truth of the trajectory, after alignment by Horn’s method [29]. In addition, the processing speed was reported as the number of frames per second processed (FPS) when the camera pose is provided.

The results obtained are shown in Figure 7. Note that the process was repeated for our method using different values of τ. Thus, each bar of the graphs shows the average values obtained in terms of ATE, pTrck and FPS for the two scenarios used (walls and ceilings).

According to the results obtained by our method, we highlight the behavior offered by our method for τ=8, the value by which our method shows its best performance and balance in terms of speed, accuracy and percentage of frames correctly tracked. For a better analysis of the results obtained, the table below Figure 7 shows a summary of the results obtained by the different methods in the two scenarios and the results obtained by our method.

In terms of ATE, our method obtained (for all values of τ) an average value of 0.03 m, which is very similar to UcoSLAM, and lower than the rest of the methods. Regarding pTrck, UcoSLAM outperforms the rest of the methods, although the sSLAM method presents an average of more than 96%. Finally, sSLAM outperforms the other FPS methods, directly related to the τ parameter. As the value of τ decreases, the number of keyframes in the map (and therefore the number of map points associated with them) decreases, which implies working with lighter maps. Note that, in any case, our method sSLAM works between 19 and 29 fps for the τ values used in the experiments, doubling the other methods’ speed.

To statistically validate the results obtained by sSLAM-8, Wilcoxon signed-rank tests were used to compare the methods. According to the results obtained in speed (FPS), there are significant differences between our method and the rest of the methods, with a significance level of 0.01. Considering the mean ATE error, our method does not show significant differences to the rest of the methods, except ORB-SLAM3, for which our method obtains better results. Finally, when we compare the pTrck, our method does not show significant differences with the rest of the methods, except for UcoSLAM and OpenVSLAM. Table 3 shows a summary of significant differences between the different methods compared. It also indicates the difference obtained between the average values of the methods compared and the Z-value obtained by the test.

In light of the results obtained, it is possible to assert that, in general, sSLAM shows a very similar accuracy to the rest of the methods in terms of ATE and percentage of successfully tracked frames, showing higher FPS.

### 5.2. Indoor Evaluation in Larger Areas

The lack of texture is one of the main problems that SLAM systems have to deal with in indoor environments. In this section, we show the performance of the vSLAM systems in an indoor area that has four corridors and an entrance that interconnects them. We placed a total of 60 hexagonal markers on the ceilings—Figure 2c—with a separation of 2 m between them, and recorded two video sequences with a mobile phone camera (with a 1920×1080 resolution) pointing toward the ceiling as it moves along the environment (see Figure 8).

Among the methods evaluated, only UcoSLAM and sSLAM are able to build the base maps; see Figure 8. The rest of the methods, using only keypoints, are incapable of finding keypoint correspondences between frames, so they end up lost. In particular, the ORB-SLAM and OpenVSLAM systems end up getting lost in the first 200 frames, making it impossible to create the map.

The base maps generated by UcoSLAM and sSLAM were employed to perform relocalization using the other video sequence. The results obtained are shown in Figure 9. Our method has a good performance in terms of ATE and pTrck for values of τ≥4. In terms of FPS, the system performance is affected by the number of keyframes used. Thus, the system shows an outstanding performance for low values of τ, obtaining its worst value when all of the keyframes are used (τ=∞).

The table of Figure 9 shows a comparison of UcoSLAM with our method for τ=8. In this case, our method presents a better performance in terms of ATE, pTrck and FPS, and it is worth noting that our method is 2.37 times faster than UcoSLAM.

### 5.3. Outdoor Analysis

This section shows the results obtained by the different vSLAM methods in outdoor areas. Unlike the previous experiments, outdoor spaces provide more textured areas, which facilitates the task of keypoint-based systems.

To perform the experiments, a total of 14 video sequences in four different areas (Zone-0, …, Zone-3) were recorded with a mobile phone at a resolution of 1920×1080 pixels (see Table 1). In each zone, a set of rectangular custom markers were uniformly placed as shown in Figure 10.

As in previous experiments, a base map was first built for each method and then used to perform SLAM with the rest of the videos. The same procedure was applied with all possible combinations of video sequences recorded in a given zone, and the average ATE, pTrck and FPS were measured.

In order to measure the ATE, it is necessary to scale and transform the trajectory to the ground truth trajectory reference system. To achieve this, a series of control points were established along the trajectory, and the Horn [29] method was applied. After the alignment of trajectories, the RMSE given between the estimated position of the camera in each frame and its actual position provided by the ground truth was evaluated.

Figure 11 shows the results obtained by the ORB-SLAM2, ORB-SLAM3, OpenVSLAM and UcoSLAM methods in the four zones evaluated. As in the previous experiments, sSLAM was evaluated for different values of τ. However, to facilitate the comparison of the methods, the table in Figure 11 shows the results obtained by our method for τ=8, for which, our method shows the best trade-off between speed and precision. The results show that our method sSLAM-8 is accurate for localization, obtaining an average value of 0.22 m, it only being outperformed by OpenVSLAM in that aspect. With regard to the speed (FPS) and pTrck, our method performs better than the other methods.

Wilcoxon signed-rank tests, with a significance level of 0.01, were applied to the results to assess the differences between the method analyzed formally. The tests indicate no significant differences in ATE and pTrck between our method and the rest of the methods, except for pTrck, where significant differences exist in favor of UcoSLAM. Concerning the speed (FPS), there are significant differences between the proposed method for all methods. Table 3 shows a summary of the significant differences in the performance of sSLAM-8 against the other methods depending on the scenario used in the experiment.

Finally, Figure 12 shows the behavior of sSLAM using τ=∞, indicating the percentage of old and new points used in the matching process. Note that using τ=∞ implies not removing keyframes for each of the 14 sequences used to perform SLAM on the base map.

To evaluate the percentage of old and new points used in the process (simulating the behavior of UcoSLAM), the base map retains all of the points. Thus, a high value in the percentage of old points would indicate that the method is using high-level points from the base map, and therefore new points are not being added or the points are not being used. In the opposite case, a high percentage of new points would indicate that the method is using, at a high level, the last points added, thus forgetting the points of the base map.

The results show that, for sequences 1–6, there is a balance between the number of new and old points; however, in the rest of the sequences there is a high percentage of new points. The reason for this behavior is that, while the first videos follow a similar trajectory in all videos, the method tends to use both points from the base map and new points. However, with non-coincident trajectories or differences in camera position, the base map points are used for relocalization. Once located, however, the system tends to follow the trajectory with the new points added. In the extreme case of low point reuse, we find sequence 7, where, although the second sequence follows the same trajectory with which it created the map, it is a very changeable environment, as shown in Figure 1.

## 6. Conclusions and Future Work

Map reusability is an important feature of vSLAM that has not been thoroughly addressed in the literature. In order to use vSLAM in real use cases, a previously created map of the environment needs to be reusable. Since environments change over time, navigation over the same environment becomes challenging when only keypoints are used.

This paper has proposed sSLAM, a new monocular SLAM method that allows for the creation of reusable maps in known environments. In the first stage, the method builds a map using keypoints and artificial markers. Once the map is finished, the keypoints are removed and only the markers are kept since we assume that they represent permanent features of the environment while most of the keypoints may change over time. After, for localization purposes, our approach relies on the markers and new temporary keypoints created along the way.

As we show in the experiments, using only the last eight keyframes is enough to obtain a tracking accuracy as high as the state-of-the-art methods, while significantly increasing the speed. As an additional advantage, sSLAM can work with customized artificial markers and operate in large indoor environments with low texture.

The proposed method was compared against state-of-the-art SLAM techniques and outperformed them in speed in all evaluated scenarios. Furthermore, the method displayed robustness, with results comparable to the best-performing methods in terms of the percentage of successfully tracked frames and absolute trajectory error.

The proposed method uses monocular cameras so it may be interesting as future work to extend it to other types of sensors, such as stereo cameras, RGBD cameras and LiDAR. Finally, it is also worth pointing out that the maps generated are camera-agnostic, in contrast to the methods that employ keypoints. Although not tested in this paper, our method can be employed for tracking with a camera different from the one employed to generate the base map.

Finally, experiments were conducted in environments where the base map was created. It would be desirable to extend this functionality in future work, allowing the base maps to be extended to new areas not explored in the first creation phase.

## Figures and Tables

**Figure 1 sensors-23-02210-f001:**
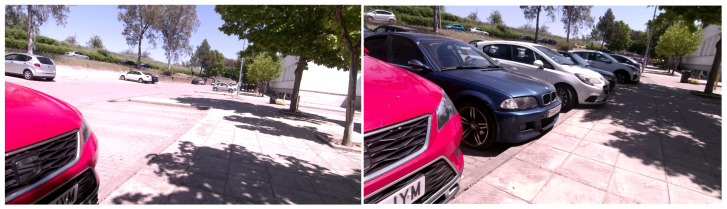
The image shows part of a scene recorded in a parking lot in the same location at two different time instants. As can be seen, the objects and shadows have changed over time, making both the relocalization process and the mapping difficult.

**Figure 2 sensors-23-02210-f002:**
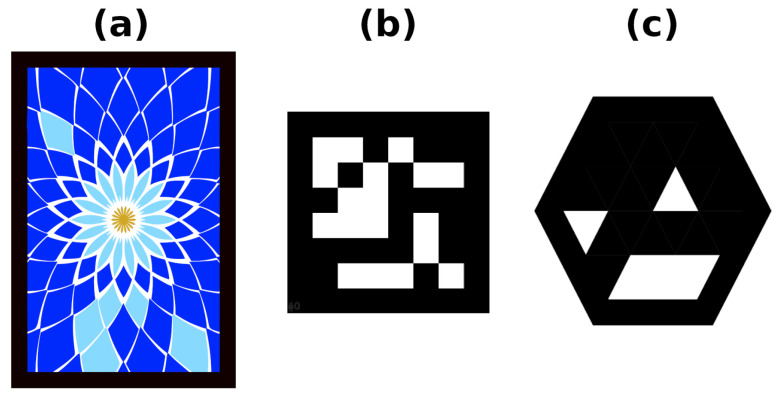
Example of markers used in this work: (**a**) rectangular color marker, (**b**) ArUco square marker, (**c**) hexagonal marker.

**Figure 3 sensors-23-02210-f003:**
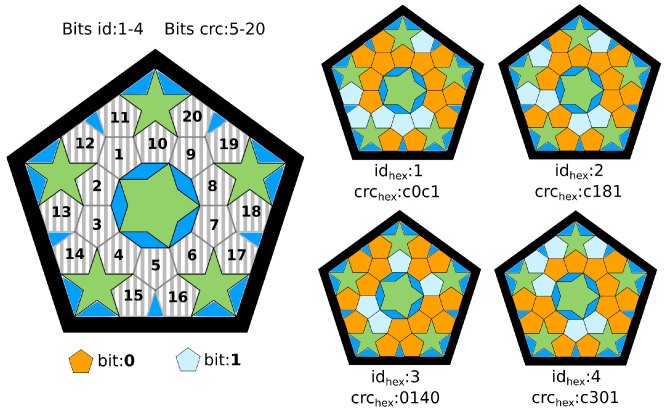
Customized markers. On the left, a custom marker template is composed of 20 bits, where bits numbered from 1–4 correspond to identification bits, and bits from 5–20 correspond to CRC bits. On the right, four different markers (and unique) are generated from the template.

**Figure 4 sensors-23-02210-f004:**
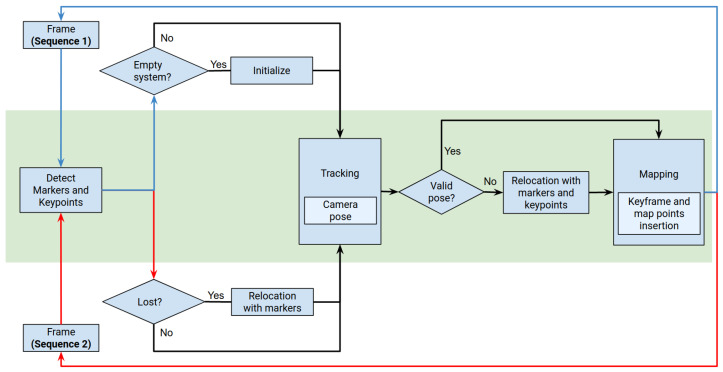
System workflow. The diagram shows the different system processes in the two phases: creation of the base map (blue lines) and localization and tracking in the previous environment (red lines). The green area shows processes shared by both phases.

**Figure 5 sensors-23-02210-f005:**
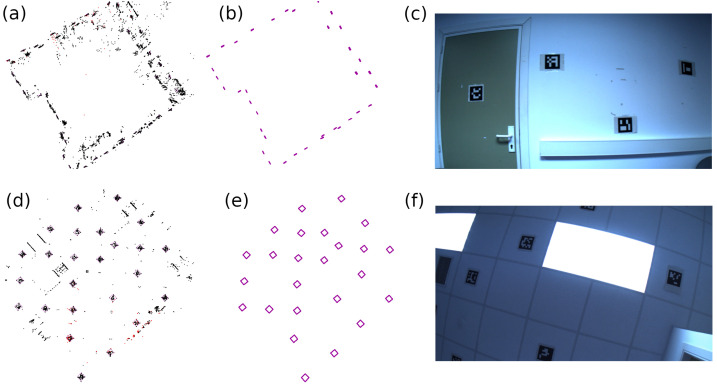
(**a**) Base map of the walls of the room built by ORB-SLAM3, composed of keypoints, and (**b**) created by sSLAM, composed only of the map of artificial markers. (**c**) Image belonging to the sequence used for the construction of the wall map. (**d**) Base map of the ceilings of the room constructed by UcoSLAM, composed of keypoints and fiducial markers, and (**e**) created by sSLAM, composed only of the map of artificial markers. (**f**) Image used for the construction of the ceiling map.

**Figure 6 sensors-23-02210-f006:**
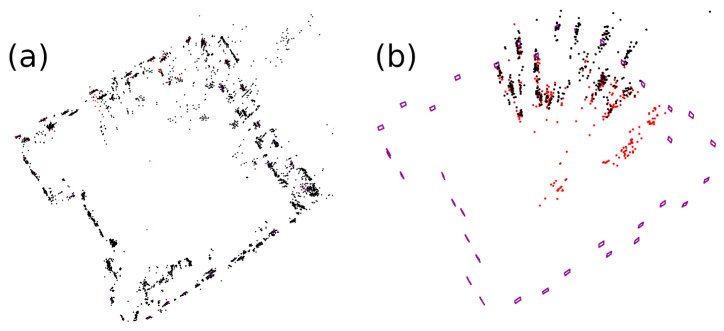
(**a**) Snapshot of the map provided by ORB-SLAM3 in the last frame of the video sequence. (**b**) The map generated by sSLAM method in the same frame, using a value of τ=10 during the SLAM process. As can be seen, our method uses the artificial markers of the base map and the keypoints of the last 10 keyframes, obtaining a lighter map.

**Figure 7 sensors-23-02210-f007:**
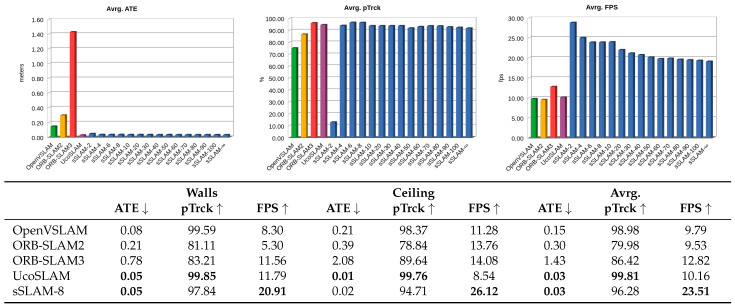
For each of the methods UcoSLAM, ORB-SLAM2, ORB-SLAM3, OpenVSLAM and sSLAM: absolute trajectory error (ATE) in meters, percentage of frames correctly tracked (pTrck) and frames per second (FPS). The graphs at the top show the overall results obtained. For sSLAM, the results of different values of τ are reported, i.e., sSLAM-8 stands for the results of sSLAM using τ=8. The table below shows the numerical results, for a comparison analysis, where the best cases are marked in bold. See text for further details.

**Figure 8 sensors-23-02210-f008:**
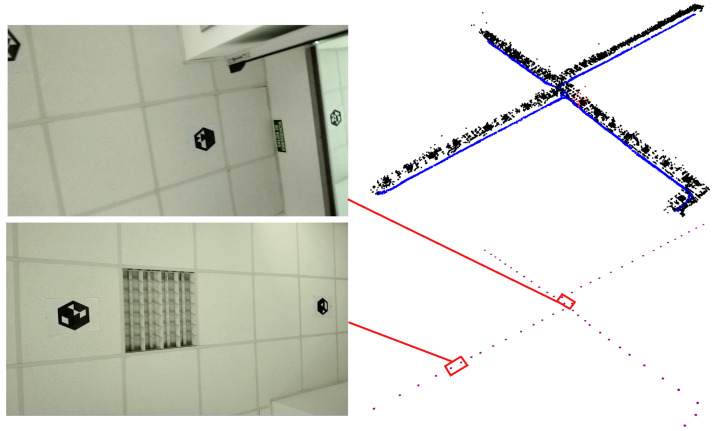
Example of the map with sSLAM using hexagonal markers in the ceiling. The **upper right** image shows the resulting map, where black points represent the three-dimensional position of the keypoints and the blue elements represent the keyframes. The **bottom right** image shows the marker map and the red elements represent the positions of the markers found. The images on the **left** show two frames of the video sequence employed.

**Figure 9 sensors-23-02210-f009:**
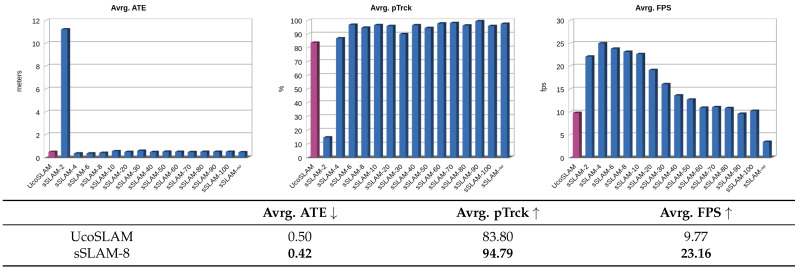
Results obtained by UcoSLAM and sSLAM in long indoor environments. We report the absolute trajectory error mean (ATE), percentage of successfully tracked frames (pTrck) and frames per second (FPS). For our method, different values of τ were tested.

**Figure 10 sensors-23-02210-f010:**
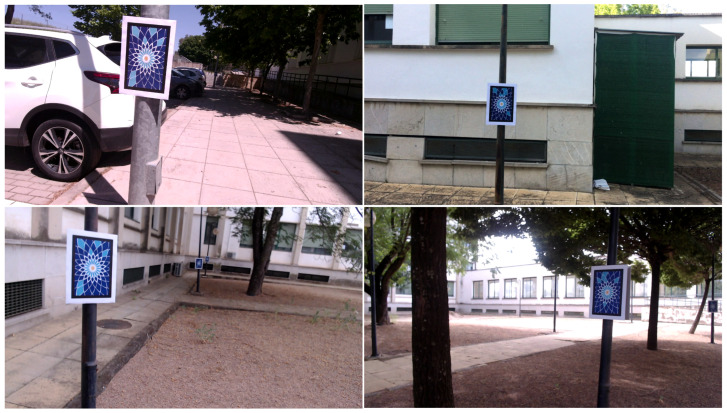
Set of example images of the outdoor environments used in the experimentation.

**Figure 11 sensors-23-02210-f011:**
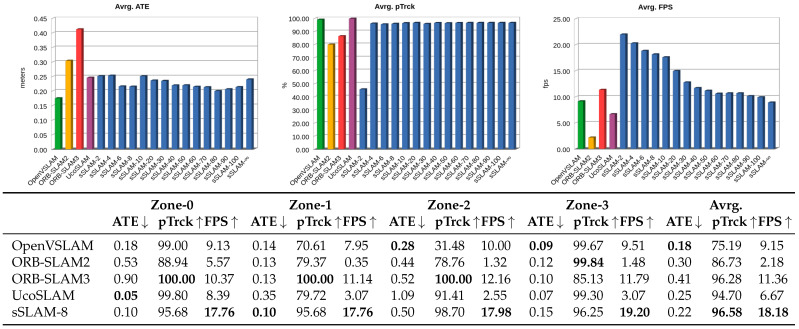
ATE, pTrck and FPS values obtained by the ORB-SLAM2, ORB-SLAM3, OpenVSLAM, UcoSLAM and sSLAM methods for the four outdoor zones.

**Figure 12 sensors-23-02210-f012:**
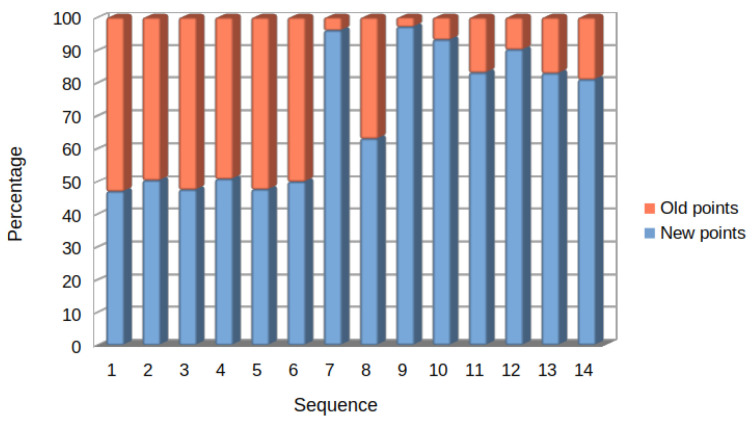
Percentage of new and old 3D points used when sSLAM performs SLAM on the base map. As can be seen, whereas in sequences 1–6, the correspondences between new and old points are balanced, the rest of the sequences show a preference for using new points, which would indicate that the old points of the map are ignored when performing SLAM.

**Table 1 sensors-23-02210-t001:** Video sequences used in the experimentation. The set of videos was used for the creation of the base map and, in a later phase, using cross-validation to produce the ATE, pTrck and FPS results.

Experiment	Zone	Video (#Frames)
Indoor room	Ceilings	video-01 (3106)
		video-02 (2603)
		video-03 (2403)
	Walls	video-04 (1682)
		video-05 (2889)
		video-06 (3458)
Large indoor	Corridor	video-07 (16,838)
		video-08 (17,420)
Outdoor	Zone-0	video-09 (7456)
		video-10 (8361)
		video-11 (6885)
		video-12 (6329)
		video-13 (6885)
	Zone-1	video-14 (1888)
		video-15 (1710)
		video-16 (1891)
	Zone-2	video-17 (6190)
		video-18 (4922)
	Zone-3	video-19 (5335)
		video-20 (5453)
		video-21 (5177)
		video-22 (5191)

**Table 2 sensors-23-02210-t002:** Set of parameters used for sSLAM in the experimentation.

Parameter	Value	Description
σ	1.2	Subsampling scale factor (Section 3.2)
η	7	Maximum level of the pyramid (Section 3.3)
τ	[10…120]	Maximum number of keyframes (Section 3.4)
$\alpha_{w}$	0.2	Minimum weight of markers. (Section 3.5)
$\alpha_{m}$	0.8	Percentage of minimum image area occupied by markers. (Section 3.5)
α	30	Minimum matches required (Section 4.2.1)
$\tau_{k}$	0.8	Threshold for adding new keyframes (Section 4.2.3)

**Table 3 sensors-23-02210-t003:** Summary of the significant differences in performance between the proposed method sSLAM (τ=8) and the rest of the methods for the different environments evaluated. Wilcoxon test was used with p=0.01. Y means that there are significant differences in favor of the proposed method. Y${}^{*}$ means significant differences in favor of the other method. The difference between the average obtained by our method and the method with which it is compared, and the Z-value obtained by the test, are shown in brackets.

		Indoor	Outdoor
OpenVSLAM	ATE	N (−0.11, −1.10)	N (0.05, −1.14)
	pTrck	Y${}^{*}$ (−2.70, −2.76)	N (8.72, −0.16)
	FPS	Y (13.73, −2.93)	Y (7.85, −3.52)
ORB-SLAM2	ATE	N (−0.32, −2.35)	N (−0.11, −0.05)
	pTrck	N (11.38, −0.86)	N (5.22, −0.67)
	FPS	Y (14.14, −3.06)	Y (14.25, −3.52)
ORB-SLAM3	ATE	Y (−1.46, −2.93)	N (−0.22, −0.05)
	pTrck	N (2.00, −1.07)	N (−2.82, −1.91)
	FPS	Y (10.58, −2.93)	Y (5.77, −2.69)
UcoSLAM	ATE	N (0.01, −2.35)	N (−0.05, −1.45)
	pTrck	Y${}^{*}$ (−3.53, −3.06)	Y${}^{*}$ (−0.42, −3.06)
	FPS	Y (13.35, −3.06)	Y (11.67, −3.06)

## Data Availability

The proposed methods and the employed data are available at: https://www.uco.es/investiga/grupos/ava/portfolio/sslam/ (Last accessed on 16 January 2023).

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
