# Peer review of "sSLAM: Speeded-Up Visual SLAM Mixing Artificial Markers and Temporary Keypoints"

_sensors, 2023, doi:10.3390/s23042210_

Round 1

Reviewer 1 Report

The paper is well written with minor points to be addressed to improve the readability and clarity for the readers:

1. A chart or diagram will be helpful to help readers understand the proposed method.

2. There is no mention on the typical camera movement speed during the experiments.

3. The sentences in line 276-280 need to be restructured.

4. Paragraph in line 530-547 can be separated to improve clarity.

5. The conclusions can be improved by stating the brief summary of performance comparison against other methods.

Author Response

Dear sir, Thanks for your time and effort in reviewing the paper. Below is a detailed response to all your suggestions. The changes in the paper have been set in blue font to ease their detection.

The paper is well written with minor points to be addressed to improve the readability and clarity for the readers:

  1. A chart or diagram will be helpful to help readers understand the proposed method.

As requested, Figure 4 has been added to adequately explain the workflow of the various parts of the system.

  1. There is no mention on the typical camera movement speed during the experiments.

As requested, the following text has been added (line 408-409):

“For the experiments, 22 video sequences were used from different zones. The camera captured the sequences while moving between $4$ and $6$ km/h in each environment.”

  1. The sentences in line 276-280 need to be restructured.

As suggested, we have restructured the paragraph as follows:

“The system initializes an empty map with the first detected keypoints and fiducial markers from a video sequence of the area to be mapped. It then continuously performs localization and mapping by adding new keypoints and markers to the map. Once the map is finished and globally optimized, all map points are discarded, and only marker information is retained.”

  1. Paragraph in line 530-547 can be separated to improve clarity.

As suggested, we separated this paragraph into two parts for better clarity.

  1. The conclusions can be improved by stating the brief summary of performance comparison against other methods.

The following text has been added (line 562-565):

“The proposed method was compared against state-of-the-art SLAM techniques and outperformed them in speed in all evaluated scenarios. Furthermore, the method displayed robustness, with results comparable to the best-performing methods in terms of the percentage of successfully tracked frames and absolute trajectory error.”

Reviewer 2 Report

The purpose of this paper is to achieve a robust and fast visual localization in environments where artificial markers have been placed. The paper has practical applications and sufficient experiments However, the method of simply eliminating key points is insufficient in terms of innovation. Consequently, I have some suggestions for the authors.

1.      In the part of the “System overview”, You can add a definition of the coordinate system.

2.      The descriptions of existing terms or definitions can be simplified.

3.      In line 410, you mentioned that the data used in the experiment were collected on your university campus. However, in line426, Is the description of “have employed the SPM dataset”? Where exactly does the first 6 indoor datasets come from?

4.      I noticed that in your experiments, you only tested the algorithm within the same environment after creating a base map, limiting its potential applications. Could you conduct experiments where the algorithm is applied to overlapping but distinct environments after the base map has been created?

Author Response

Dear sir, Thanks for your time and effort in reviewing the paper. Below is a detailed response to all your suggestions. The changes in the paper have been set in blue font to ease their detection.

The purpose of this paper is to achieve a robust and fast visual localization in environments where artificial markers have been placed. The paper has practical applications and sufficient experiments However, the method of simply eliminating key points is insufficient in terms of innovation. Consequently, I have some suggestions for the authors.

  1. In the part of the “System overview”, You can add a definition of the coordinate system.

As suggested, the following text has been added (line 192-193):

“Note that a reference system describes objects' relative positions (location and orientation) in a specific space.”

  1. The descriptions of existing terms or definitions can be simplified.

It is unclear which terms or definitions the reviewer refers to, so we are unsure what changes to make to the paper. In any case, our definitions aim at being easy to understand by any reader, so we consider keeping them as they are not harmful. 

  1. In line 410, you mentioned that the data used in the experiment were collected on your university campus. However, in line426, Is the description of “have employed the SPM dataset”? Where exactly does the first 6 indoor datasets come from?

All datasets have been recorded at our University Campus. While the corridors and outdoor areas have been recorded for this work, indoor areas (corridors and ceiling) correspond to a laboratory area already created in our previous work [6].

For clarity, the text has been modified (line 408-412):

“For the experiments, 22 video sequences were used from different zones. The camera captured the sequences while moving between $4$ and $6$ km/h in each environment. The first 6 sequences belonged to the SPM dataset \cite{SPMSLAM} and were recorded indoors, spotting at walls and ceilings. The remaining 16 sequences include 2 from the corridors and lobby of a building and 14 from various outdoor locations on our University Campus, see Table “

  1. I noticed that in your experiments, you only tested the algorithm within the same environment after creating a base map, limiting its potential applications. Could you conduct experiments where the algorithm is applied to overlapping but distinct environments after the base map has been created?

The reviewer is correct. Our work mainly focuses on creating maps of areas where an autonomous agent (e.g., a robot) will eventually move to do a task. This work is not focused on exploring unknown areas while doing the tracking task since it is dangerous in many use cases. Nevertheless, our recordings include different viewpoints on the same area, thus exploring different paths on the map. 

As future work, we find it interesting to expand the base map to incorporate previously unexplored areas.

This sentence has been added to the text (line 572-574):

“Finally, the experiments have been conducted in environments where the base map was created. As future work, it would be desirable to extend this functionality, allowing the base maps to be extended to new areas not explored in the first phase.”
